# SELF-ALIGNMENT LEARNING TO IMPROVE MYOCARDIAL INFARCTION DETECTION FROM SINGLE-LEAD ECG

## ABSTRACT

Myocardial infarction is a critical manifestation of coronary artery disease, yet detecting it from single-lead electrocardiogram (ECG) remains challenging due to limited spatial information. An intuitive idea is to convert single-lead into multiple-lead ECG for classification by pre-trained models, but generative methods optimized at the signal level in most cases leave a large latent space gap, ultimately degrading diagnostic performance. This naturally raises the question of whether latent space alignment could help. However, most prior ECG alignment methods focus on learning transformation invariance, which mismatches the goal of single-lead detection. To address this issue, we propose SelfMIS, a simple yet effective alignment learning framework to improve myocardial infarction detection from single-lead ECG. Discarding manual data augmentations, SelfMIS employs a self-cutting strategy to pair multiple-lead ECG with their corresponding single-lead segments and directly align them in the latent space. This design shifts the learning objective from pursuing transformation invariance to enriching the single-lead representation, explicitly driving the single-lead ECG encoder to learn a representation capable of inferring global cardiac context from the local signal. Experimentally, SelfMIS achieves superior performance over baseline models across nine myocardial infarction types while maintaining a simpler architecture and lower computational overhead, thereby substantiating the efficacy of direct latent space alignment. Our code and checkpoint will be publicly available after acceptance.

## 1 INTRODUCTION

Myocardial infarction, a critical manifestation of coronary artery disease, is a leading cause of mortality worldwide (Reed et al., 2017). It is responsible for over 2.4 million deaths in the United States and 4 million in Europe and North Asia (Nichols et al., 2014), representing over one-third of all annual deaths in developed nations (Yeh et al., 2010). The electrocardiogram (ECG) is a standard diagnostic tool for myocardial infarction in clinical practice, valued for its simplicity and non-invasive nature Saleh & Ambrose (2018). Conventional myocardial infarction detection methods predominantly rely on twelve-lead (hereafter referred to as multiple-lead) ECG signals. However, the requirement for specialized equipment and trained personnel, compounded by the sudden onset of myocardial infarction, severely restricts the use of multiple-lead ECG for effective screening in out-of-hospital settings (Hong et al., 2020). Consequently, many patients fail to receive a timely diagnosis within the critical "golden hour" for myocardial ischemia, thus missing the optimal opportunity for reperfusion therapy.

In recent years, the collection of single-lead ECG through mobile devices such as the Apple Watch has provided a new solution for out-of-hospital screening for cardiac diseases (Butler et al., 2024). Nonetheless, accurately detecting myocardial infarction from a single-lead ECG (e.g., Lead I) remains a formidable challenge. This difficulty is rooted in the inherent single-view limitation of the signal, which captures only one projection of the heart's complete electrical activity (Han et al., 2024). Consequently, supervised learning methods exhibit limited sensitivity to myocardial infarction lesions in specific myocardial regions, such as the anterior or posterior walls. To enhance the diagnostic performance of single-lead ECG, an intuitive idea is to convert single-lead ECG into

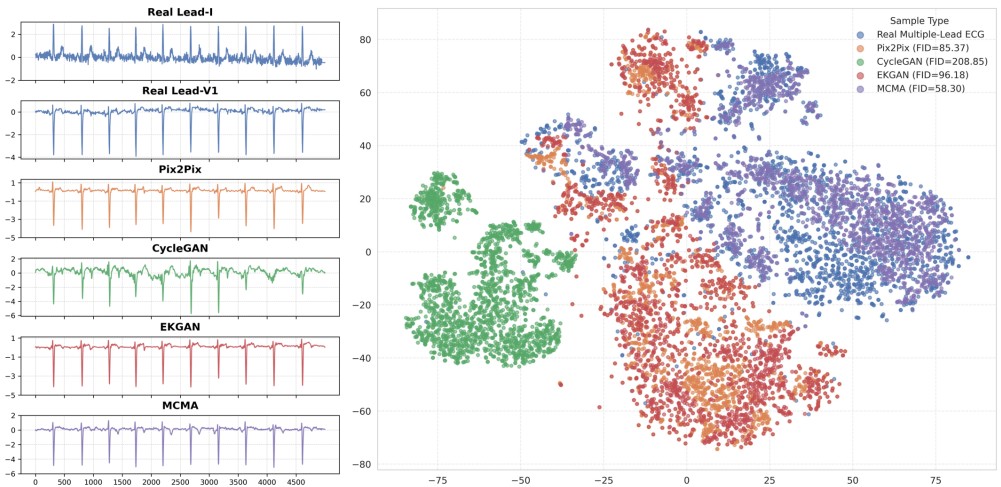

Figure 1: Looks good at the signal level? But a large gap in the latent space for the generative methods.

multiple-lead ECG and then apply a pre-trained multiple-lead ECG classifier. Such generative approaches have been widely explored in prior studies(Chen et al., 2024). However, these methods still suffer from the following limitations:

**Large Gap in Latent Space For ECG Generative methods.** Our observations indicate that although generative models can synthesize samples that appear consistent at the signal level, the gap between generated and real samples remains significant in the latent space, as illustrated in Figure 1. For example, the left panel of Figure 1 shows the generation of lead V1 based on lead I. While most methods can produce visually similar signals, when the generated and real multiple-lead samples are fed into a pre-trained multiple-lead ECG encoder, their latent representations exhibit a large discrepancy. This mismatch ultimately causes the generated samples to misalign with the discriminative feature space of the pre-trained classifier, resulting in severe performance degradation.

We attribute this issue to the fact that existing generative methods are typically optimized at the signal level, as their evaluations predominantly focus on signal-level metrics such as mean squared error (MSE). As a result, these ECG generation models generally focus only on signal-level consistency while neglecting alignment in the latent space. This naturally raises a question: if aligning signals at the signal level is insufficient, could latent space alignment methods be leveraged to address this limitation? Unfortunately, most prior alignment methods have almost exclusively focused on transformation-invariant alignment, where different augmented views of the same ECG are aligned to enforce robustness against perturbations. Such approaches also come with inherent limitations:

**Mismatch of Invariance Learning and Single-Lead Detection.** The core objective of prior ECG alignment methods is to learn invariance to transformations, i.e., to ensure robustness under signal perturbations, as illustrated in Figure 2 (b). However, the key challenge in single-lead myocardial infarction detection is different: the model must infer the global cardiac state from a constrained local projection, including disease-related features that are not directly observable in a single-lead. Since previous alignment pre-training is not designed to cultivate this extrapolation ability, the learned representations, though robust, lack the necessary cross-lead contextual information. Moreover, for ECG signals, commonly used simple data augmentations may severely distort the original semantic information, thereby degrading alignment performance(Lan et al., 2024). This mismatch between the learning objective and downstream requirements results in a pronounced performance drop for single-lead ECG myocardial infarction detection.

To address these limitations, we introduce **Self**-alignment learning to improve **M**yocardial **I**nfarction detection from **S**ingle-lead ECG (**SelfMIS**). Without relying on intricate architectural designs, Self-MIS is built upon a straightforward principle: by employing a self-cutting strategy to pair multiple-lead ECG with their corresponding single-lead segments and directly aligning them in the latent space. SelfMIS discards manual data augmentations and adopts a self-cutting strategy for construct-

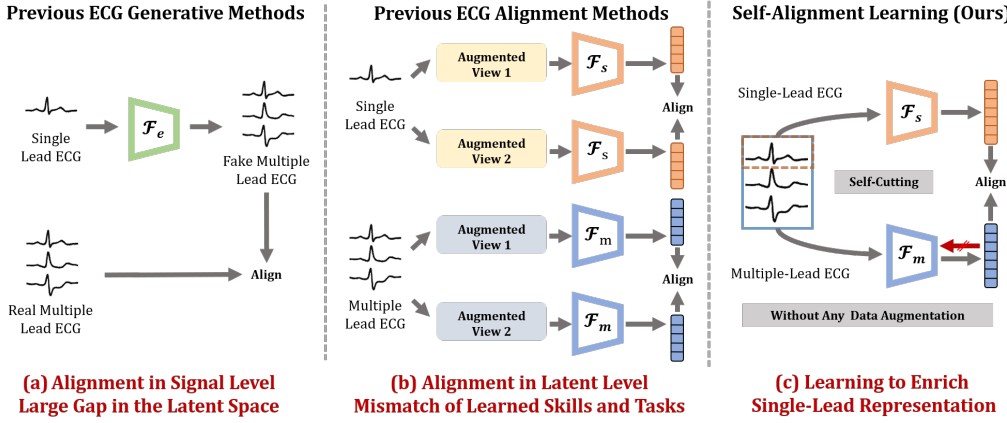

Figure 2: **(a)** Previous generative methods align only at the signal level, leaving a large latent space gap. **(b)** Previous alignment methods rely on augmented single-lead or multiple-lead ECG to learn invariance, which mismatches single-lead detection. **(c)** Our SelfMIS directly aligns single-lead with multiple-lead ECG in the latent space via self-cutting, enriching single-lead representations for improved detection.

ing positive pairs, where "self" indicates that the aligned objects are entirely derived from the original sample itself rather than generated through data augmentation. During the alignment process, we apply a stop-gradient operation to the multiple-lead ECG encoder, thereby forcing the single-lead ECG encoder to align its representations with the discriminative space of the multiple-lead ECG encoder. Consequently, the learning objective shifts from enforcing transformation invariance to fostering representation enrichment. Our framework explicitly drives the single-lead ECG encoder to capture global cardiac context from local signals, producing information enriched embeddings that substantially enhance myocardial infarction detection performance. The main contributions of this work are summarized in below:

- We propose SelfMIS, a simple yet effective alignment learning framework to improve myocardial infarction detection from single-lead ECG that achieves superior performance over baseline models while maintaining architectural simplicity and computational efficiency.

- We introduce a self-cutting based positive-pair construction strategy that abandons manual data augmentation, thereby providing a concise, augmentation-free paradigm that preserves complete semantics during the alignment process.

- We present a straightforward latent space alignment strategy that freezes the multiple-lead ECG encoder to directly align single-lead ECG representations with the discriminative space of the multiple-lead ECG, thereby enriching the single-lead representations.

- To facilitate future research, we establish the first benchmark for single-lead myocardial infarction detection by comparing SelfMIS against 8 discriminative and 4 generative methods across 9 myocardial infarction types. Experimental results demonstrate that SelfMIS achieves a macro AUC score of over 70.0 for each of the nine infarction types and an overall average macro AUC exceeding 80.0, showcasing its promising performance.

## 2 METHODOLOGY

### 2.1 OVERVIEW OF SELFMIS

SelfMIS performs representation learning by minimizing the distance between single-lead and multiple-lead ECG representations in the latent space, as illustrated in Figure 3. Instead of relying on augmented views, SelfMIS directly leverages the inherent physiological relationship between single-lead and multiple-lead ECG to construct positive pairs. Consequently, the learning objective

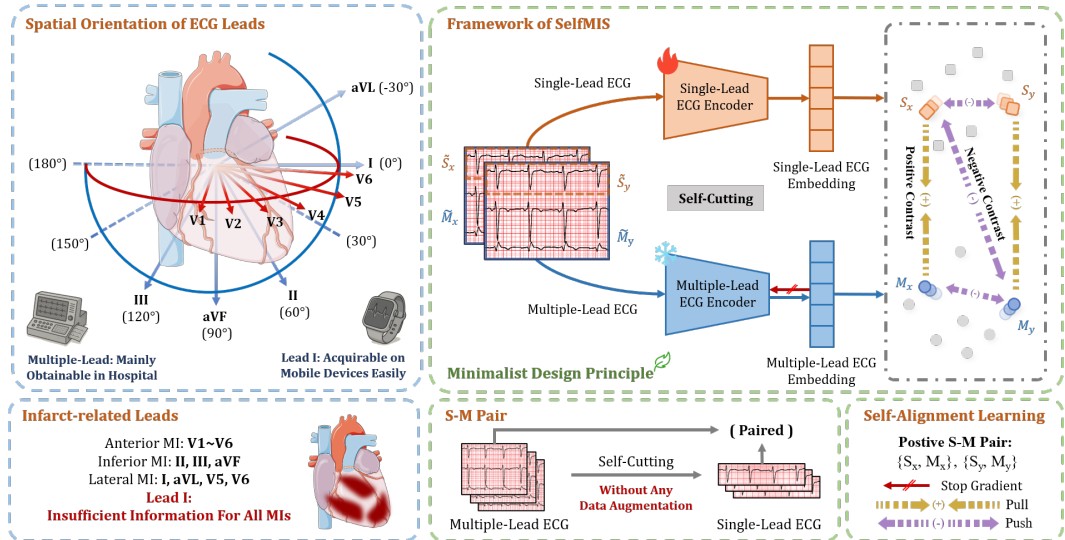

Figure 3: The limited view of single-lead ECG hinders their ability to capture complete myocardial infarction features. SelfMIS mitigates this by constructing positive S–M pairs through self-cutting and aligning their representations in latent space. During training, gradients to the multiple-lead ECG encoder are stopped, forcing the single-lead ECG encoder to adapt to the discriminative feature space of the multiple-lead ECG.

shifts from enforcing invariance across augmentations to guiding the single-lead ECG encoder to capture global cardiac context from local signals. The design of SelfMIS adheres to the principle of minimalism, with all components implemented in their simplest form, thereby demonstrating the effectiveness of our paradigm without relying on additional tricks. SelfMIS is primarily composed of the following components:

## 2.2 POSITIVE PAIR CONSTRUCTION

The construction of positive pairs in SelfMIS is straightforward. Specifically, given a multiple-lead ECG $\tilde{M} \in \mathbb{R}^{c \times t}$, where $c$ denotes the number of leads and $t$ represents the total number of timestamps, we employ a "self-cutting" approach to directly extract the ECG signal from Lead I, $\tilde{S} \in \mathbb{R}^{1 \times t}$, without any additional data augmentation. We refer to the positive pair as an S-M pair, which is formed by pairing the multiple-lead signal with its corresponding single-lead signal from the same ECG record $i$, formulated as:

$$P_i^+ := (\tilde{S}_i, \tilde{M}_i). \tag{1}$$

## 2.3 BACKBONE NETWORK ARCHITECTURE

SelfMIS imposes no restrictions on the choice of backbone network. In our implementation, we select a standard ResNet1D for its simplicity and efficiency. A pair of encoders, a single-lead ResNet1D and a multiple-lead ResNet1D, are used to extract embeddings from the positive pair $(\tilde{S}_i, \tilde{M}_i)$. Crucially, the gradient flow to the multiple-lead ECG encoder is detached during pre-training, compelling the single-lead representation to align with the discriminative feature space of its multiple-lead counterpart. Furthermore, since ECGFounder (Li et al., 2025) utilizes the same architecture, we leverage its pre-trained weights to bootstrap the representation alignment process. Let $f_s(\cdot)$ and $f_m(\cdot)$ denote the single-lead and multiple-lead ECG encoders, respectively. We obtain the corresponding latent representations as:

$$S_i = f_s(\tilde{S}_i), \quad M_i = \text{stopgrad}\big(f_m(\tilde{M}_i)\big), \tag{2}$$

## 2.4 Loss Function

SelfMIS adopts a simple SigLIP loss (Zhai et al., 2023) for training. Unlike alignment losses that rely on softmax normalization across the batch, SigLIP independently processes each S-M pair and formulates the problem as a standard binary classification task over all possible combinations. Concretely, given a batch $\mathcal{B}$ of $|\mathcal{B}|$ pairs, the label $z_{ij} = 1$ if $(S_i, M_j)$ is a matched pair and $z_{ij} = -1$ otherwise. Here, negative pairs are naturally constructed by pairing each single-lead ECG with all non-matching multiple-lead ECG within the same batch. The loss is defined as:

$$\mathcal{L} = -\frac{1}{|\mathcal{B}|} \sum_{i=1}^{|\mathcal{B}|} \sum_{j=1}^{|\mathcal{B}|} \log \left( \frac{1}{1 + \exp\left(z_{ij} \cdot (-t\langle S_i, M_j\rangle + b)\right)} \right), \tag{3}$$

where $\langle S_i, M_j\rangle$ denotes the inner product between the single-lead and multiple-lead embeddings, $t$ is a learnable temperature parameter, and $b$ is a bias term. For simplicity, the bias term $b$ is fixed to zero, and the temperature $t$ is set to $0.07$. Intuitively, this loss encourages the embeddings of positive pairs to move closer, thereby aligning the single-lead ECG encoder with the discriminative feature space of the multiple-lead ECG encoder. As a result, the single-lead ECG encoder acquires richer and more clinically informative representations, ultimately enhancing the performance of myocardial infarction detection from single-lead ECG.

## 3 Experiments

### 3.1 Pre-training Configuration

**MIMIC-IV-ECG.** This publicly available dataset (Gow et al., 2023) provides a comprehensive collection of 800,035 multiple-lead ECG recordings from 161,352 subjects. Each recording was sampled at a rate of 500 Hz and has a duration of 10 seconds. For data preprocessing, we addressed missing values by replacing "NaN" and "Inf" entries with zero and standardized the lead order of each ECG recordings.

**Implementation.** Pre-training was performed using the original 500 Hz sampling rate, while the final 5,000 samples of the dataset were designated as a validation set for computing retrieval metrics during validation stage. We trained the model for 20 epochs with the Adam optimizer, using an initial learning rate of 0.0001 and a cosine annealing scheduler. The pre-trained weights of ECGFounder were loaded for both the single-lead ECG and multiple-lead ECG encoders for SelfMIS pre-training, and the multiple-lead ECG encoder was frozen. All experiments were conducted on four NVIDIA RTX 4090 GPUs, with a batch size of 128 per GPU.

### 3.2 Downstream Tasks Configuration

**PTB-XL.** This publicly accessible dataset (Wagner et al., 2020) consists of 21,837 multiple-lead ECG recordings collected from 18,885 patients. Each ECG signal was sampled at 500 Hz and has a fixed duration of 10 seconds. Based on the SCP-ECG protocol, the multi-label classification task has several subsets. We primarily focused on the diagnostic subset, which includes a total of 44 classes, covering 9 types of myocardial infarction diseases. The full names corresponding to the abbreviations for the nine types of myocardial infarction are detailed in the *appendix*. We followed the official data split (Strodthoff et al., 2021) for training, validation and testing.

**Implementation.** Fine-tuning was performed using only the data from lead I. In cases where a baseline method was incompatible with single-lead input, we adapted the input by applying a zero-mask to the multiple-lead signal, ensuring that only the data from lead I was preserved. We maintained the original 500 Hz sampling rate for fine-tuning and applied z-score normalization to all downstream datasets. For linear probing, we kept the encoder frozen, updating only the parameters of a randomly initialized linear classifier. We trained the single-lead ECG model for 20 epochs with the Adam optimizer with an early stopping patience of 5, using an initial learning rate of 0.0005 and a cosine annealing scheduler. It is important to note that all final test results were derived from the model that achieved the best performance on the validation set, avoiding the practice of reporting the highest test result across all epochs. The macro ROC-AUC (hereafter referred to as macro AUC) was used as the primary metric for all downstream tasks. Further experimental details are provided in the *appendix*.

Table 1: Linear probing results of SelfMIS and other discriminative methods. The best results are **bolded**, with gray indicating the second highest.

| Method | ALMI | AMI | ASMI | ILMI | IMI | IPLMI | IPMI | LMI | PMI | Avg |
|---|---|---|---|---|---|---|---|---|---|---|
| SimCLR (Chen et al., 2020) | 59.71 | 52.99 | 74.19 | 61.50 | 61.49 | 60.46 | 67.18 | 52.18 | 56.45 | 60.68 |
| Wav2Vec 2 (Baevski et al., 2020) | 69.53 | 53.94 | 68.59 | 57.82 | 56.20 | 66.56 | 77.28 | 68.93 | 51.74 | 63.40 |
| 3KG (Gopal et al., 2021) | 67.70 | 55.62 | 74.35 | 64.59 | 58.94 | 73.00 | 51.17 | 66.05 | 46.22 | 61.96 |
| CLOCS (Kiyasseh et al., 2021) | 84.79 | 66.66 | 74.03 | 68.06 | 59.48 | 45.16 | 76.74 | 71.12 | 47.26 | 65.92 |
| W2V-CMSC (Oh et al., 2022) | 92.12 | 71.63 | 82.42 | **80.37** | 65.88 | 61.59 | 73.70 | 81.02 | 95.15 | 78.21 |
| ST-MEM (Na et al., 2024) | 69.60 | 52.16 | **86.35** | 79.69 | 69.76 | 72.22 | 74.09 | 68.41 | 87.50 | 73.31 |
| HeartLang (Jin et al., 2025) | 87.56 | 66.86 | 81.84 | 72.67 | 66.27 | 70.18 | 63.06 | **85.03** | 91.54 | 76.11 |
| ECGFounder (Li et al., 2025) | 86.11 | 68.09 | 81.68 | 72.94 | 66.37 | 58.52 | 78.67 | 80.69 | 73.21 | 74.03 |
| SelfMIS (Ours) | **92.96** | **72.81** | 84.81 | 79.37 | **71.19** | **75.06** | **84.41** | 83.78 | **98.56** | **82.55** |

Table 2: Linear probing results of SelfMIS and other generative methods using the multiple-lead ECGFounder as the classifier. The best results are **bolded**, with gray indicating the second highest.

| Method | ALMI | AMI | ASMI | ILMI | IMI | IPLMI | IPMI | LMI | PMI | Avg |
|---|---|---|---|---|---|---|---|---|---|---|
| Pix2Pix (Isola et al., 2017) | 70.29 | 57.37 | 55.73 | 58.92 | 52.29 | **76.11** | 63.57 | 63.11 | 84.30 | 64.63 |
| CycleGAN (Zhu et al., 2017) | 61.51 | 52.63 | 52.84 | 55.02 | 46.88 | 41.68 | 52.34 | 65.23 | 70.69 | 55.42 |
| EKGAN (Joo et al., 2023) | 57.91 | 63.04 | 47.46 | 54.20 | 51.89 | 68.24 | 59.92 | 75.66 | 65.31 | 60.40 |
| MCMA (Chen et al., 2024) | 91.01 | 70.97 | 80.17 | 70.79 | 65.37 | 68.54 | 67.42 | 74.33 | 97.24 | 76.20 |
| SelfMIS (Ours) | **92.96** | **72.81** | **84.81** | **79.37** | **71.19** | 75.06 | **84.41** | **83.78** | **98.56** | **82.55** |

## 4 RESULTS AND DISCUSSIONS

### 4.1 EVALUATION RESULTS COMPARED WITH DISCRIMINATIVE METHODS

**Results.** Table 1 shows the linear probing results of SelfMIS in comparison with existing discriminative methods. Our proposed method SelfMIS exhibits a clear advantage across the nine myocardial infarction diseases. In particular, SelfMIS achieves a substantial overall average improvement of 4.34 in macro AUC performance over W2V-CMSC, which is the second-best method on average. When compared to ECGFounder, which utilizes an identical network architecture, SelfMIS improves the overall average macro AUC performance by 8.52. A noteworthy observation is that SelfMIS consistently achieves a macro AUC above 70.0 for all individual myocardial infarction diseases, with its overall average macro AUC exceeding 80.0, a performance level that other methods fail to achieve. These results effectively demonstrate the significant superiority of our proposed method SelfMIS for single-lead myocardial infarction detection.

**Discussions.** We posit that the suboptimal performance of different methods can be attributed to various factors. ECGFounder, a representative supervised learning approach, was pre-trained on a massive dataset HEEDB (Koscova et al., 2025) of over 10 million single-lead ECG signals. Despite having the largest pre-training dataset, its performance did not significantly improve compared to other methods. This observation indicates that the intrinsic limitations of lead I prevent it from capturing information related to all myocardial infarction, thereby establishing a performance ceiling for single-lead ECG detection by supervised learning that cannot be surpassed by simply scaling up the pre-training dataset size. For reconstruction-based self-supervised methods like ST-MEM, the ECG representations learned from the reconstruction task lack discriminative and high-level semantic information, leading to poor performance on downstream tasks. For contrastive self-supervised methods like 3KG, data augmentation techniques such as rotation, shifting, and noise injection can compromise the semantic consistency of the ECG signal, thereby degrading the performance of representation learning. Different from alignment methods that aim to learn transformation invariance, SelfMIS directly aligns the embeddings of single-lead and multiple-lead ECG within the latent space. This direct alignment mechanism effectively sidesteps the semantic distortion introduced by

Table 3: Linear probing results of ECGFounder and SelfMIS across different fine-tuning data ratios, while improved values are marked in **green**.

| Fine-tuning Data Ratio | 25% | | 50% | | 100% | |
|---|---|---|---|---|---|---|
| | ECG Founder | SelfMIS | ECG Founder | SelfMIS | ECG Founder | SelfMIS |
| ALMI | 81.91 | 92.65 | 85.14 | 91.71 | 86.11 | 92.96 |
| AMI | 62.50 | 67.47 | 68.01 | 71.09 | 68.09 | 72.81 |
| ASMI | 79.16 | 83.14 | 81.11 | 84.20 | 81.68 | 84.81 |
| ILMI | 62.84 | 72.51 | 66.25 | 76.89 | 72.94 | 79.37 |
| IMI | 63.86 | 68.44 | 65.97 | 70.02 | 66.37 | 71.19 |
| IPLMI | 50.77 | 68.32 | 62.39 | 74.97 | 58.52 | 75.06 |
| IPMI | 61.58 | 81.84 | 58.05 | 79.69 | 78.67 | 84.41 |
| LMI | 64.46 | 79.47 | 76.74 | 82.07 | 80.69 | 83.78 |
| PMI | 24.74 | 45.41 | 60.37 | 97.01 | 73.21 | 98.56 |
| Avg | 61.31 | 73.25 | 69.33 | 80.88 | 74.03 | 82.55 |
| **Avg Improvement** | **11.94↑** | | **11.54↑** | | **8.52↑** | |

data transformations and guarantees semantic consistency between the single-lead and multiple-lead representations, thereby substantially enhancing the discriminative performance on single-lead data. In summary, SelfMIS exhibits robust performance for single-lead myocardial infarction detection.

## 4.2 Evaluation Results Compared with Generative Methods

**Results.** Table 2 details the myocardial infarction detection results for SelfMIS and existing generative methods. The evaluation pipeline for the generative methods is a multi-step process: it involves using real multiple-lead training data to fine-tune multiple-lead ECGFounder while simultaneously training the generative model to map single-lead inputs to multiple-lead outputs. Subsequently, single-lead test data is passed through the generative model to produce synthetic multiple-lead data, which is then fed into the fine-tuned multiple-lead ECGFounder for final results. The effectiveness of this intricate pipeline is highly dependent on the quality of the generated data. Remarkably, Self-MIS achieves superior performance with a significantly more straightforward pipeline. Our method surpasses the second-best method MCMA, with an average macro AUC improvement of 6.35, and achieves a substantial average gain of 22.15 in macro AUC over EKGAN.

**Discussions.** We delved deeper into the reasons behind the suboptimal performance of current generative methods. Figure 1 shows the latent space distributions of the multiple-lead signals generated by these methods and the corresponding real signals. A significant discrepancy between the generated and real distributions is observed, as indicated by the high Fréchet Inception Distance (FID). This issue arises because most evaluation metrics focus on signal-level fidelity, such as mean squared error (MSE), which in turn drives existing generative methods to optimize predominantly at the signal level. As a result, while these methods can produce visually plausible samples, the latent space distribution of the generated signals remains considerably misaligned with that of real samples, which in turn degrades the performance of the downstream classifier. This finding suggests that when designing generative methods, the community should not only focus on signal-level losses but also align the generated distribution with the real distribution in latent-level to achieve better performance on downstream tasks. In summary, SelfMIS achieves superior performance despite its more straightforward pipeline.

## 4.3 Evaluation Results Across Different Fine-tuning Data Ratios

**Results.** Table 3 presents the myocardial infarction detection results of SelfMIS and ECGFounder under varying fine-tuning data sizes. We conducted experiments by randomly sampling different percentages of the training set data, while the validation and test sets remained unchanged. SelfMIS demonstrates a significant advantage regardless of the amount of fine-tuning data. Specifically, SelfMIS achieves an average macro AUC performance improvement of 11.94, 11.54, and 8.52 over ECGFounder with 25%, 50%, and 100% of the training data, respectively. Notably, even with only

Table 4: Pre-training resource usage results of SelfMIS and other methods, while "Same Batch Size" is set to 128, and "Default Config" refers to the batch size in the original code configuration file.

| Method | Parameters | FLOPs | GPU Memory Usage | |
|---|---|---|---|---|
| | | | Same Batch Size | Default Config |
| SimCLR (Chen et al., 2020) | 90.37 M | 18.06 GMac | 59548 MiB | 30826 MiB |
| Wav2Vec 2 (Baevski et al., 2020) | 90.88 M | 18.13 GMac | 63282 MiB | 74882 MiB |
| 3KG (Gopal et al., 2021) | 90.37 M | 18.06 GMac | 59548 MiB | 30826 MiB |
| CLOCS (Kiyasseh et al., 2021) | 90.37 M | 18.06 GMac | 31254 MiB | 16560 MiB |
| W2V-CMSC (Oh et al., 2022) | 90.88 M | 18.13 GMac | 79592 MiB | 34446 MiB |
| ST-MEM (Na et al., 2024) | 88.5 M | 36.79 GMac | 10318 MiB | 2952MiB |
| HeartLang (Jin et al., 2025) | 44.56 M | 10.6 GMac | 18562 MiB | 9868 MiB |
| ECGFounder (Li et al., 2025) | 30.81 M | 1.15 GMac | - | - |
| SelfMIS (w/o AMP) | 61.32 M | 2.33 GMac | 10404 MiB | 10404 MiB |
| SelfMIS (w AMP) | 61.32 M | 2.33 GMac | 5926 MiB | 5926 MiB |

25% of the training data, SelfMIS still achieves a macro AUC level greater than 70.0 for most myocardial infarction diseases, closely approaching ECGFounder's performance with 100% of the data. Furthermore, with just 50% of the training data, SelfMIS consistently achieves a detection performance greater than 70.0 in macro AUC for all myocardial infarction diseases, with the overall average surpassing 80.0.

**Discussions.** Acquiring large-scale labeled data is costly and time-consuming in real-world clinical applications. These results demonstrate that SelfMIS owes its strength in low-data settings to the latent space alignment strategy. In contrast, ECGFounder, which relies purely on supervised learning under the same architecture, suffers pronounced performance degradation and even loses diagnostic utility on some diseases when trained with merely 25% of the data. In contrast, SelfMIS can still effectively diagnose the majority of myocardial infarction diseases, with the exceptions of PMI and IPLMI. Although the performance gain of SelfMIS over ECGFounder decreases when using the full 100% of the data, this does not undermine its strength. This trend may indicate that with an ample amount of data, the performance of both models tends to converge. In summary, SelfMIS exhibits stronger generalization capabilities under data-limited conditions.

### 4.4 PRE-TRAINING RESOURCE USAGE COMPARISON

**Results.** Table 4 presents a comparison of the pre-training resource consumption of SelfMIS and other discriminative methods. We conducted these experiments using the well-known open-source library fairseq-signals[1] or the corresponding open-source code for each method. The SelfMIS model demonstrates superior efficiency across parameter count, computational complexity, and GPU memory usage. Since the training code of ECGFounder has not been released, GPU memory consumption is temporarily not included. Specifically, the FLOPs of SelfMIS is 2.33 GMac, which is approximately one-tenth of other self-supervised learning methods, and its parameter count of 61.32 M is also lower. For a fair comparison of GPU memory usage, we standardized the batch size to 128 for all methods, as they originally used different batch sizes. Under this unified setting, our method achieved a GPU memory footprint of $10,404$ MiB without automatic mixed precision (AMP) and reached a minimum usage of $5,926$ MiB with AMP enabled.

**Discussions.** Pretraining resource consumption has often been neglected in prior research. Although performance metrics such as macro AUC are essential for assessing model quality, they alone fail to provide a comprehensive evaluation. The practical viability of a model, especially in resource-constrained environments, is equally dependent on its computational cost. Lower FLOPs signify a reduced computational load per operation, which directly leads to faster inference, lower power consumption, and less demanding hardware specifications. Based on its simple yet effective positive pair definition, S-M pair alignment, and mixed-precision computing, SelfMIS achieves very low computational resource usage, particularly concerning GPU memory. Even without activating AMP,

---

[1]https://github.com/Jwoo5/fairseq-signals

Table 5: Ablation study for our proposed method SelfMIS, where "S" denotes single-lead ECG encoder and "M" denotes multiple-lead ECG encoder.

| Ablation Options | 1-Lead to 12-Lead Retrieval | | | | Avg AUC |
|---|---|---|---|---|---|
| | R@1 | R@5 | R@10 | Loss | |
| w/o Alignment | - | - | - | - | 65.85 |
| w/o S Pre-trained Checkpoint | 34.10 | 55.54 | 64.24 | 1.2024 | 79.54 |
| w/o M Pre-trained Checkpoint | 11.94 | 30.44 | 41.52 | 1.8946 | 76.71 |
| w/o S-M Pre-trained Checkpoint | 8.68 | 22.30 | 28.78 | 2.9036 | 76.56 |
| w S-M Pre-trained Checkpoint | 45.98 | 74.08 | 82.60 | 0.6274 | 82.55 |

the FLOPs of SelfMIS are substantially lower than ST-MEM which with a comparable GPU memory footprint, enabling the more rapid pre-training and fine-tuning processes. In summary, SelfMIS is more computationally efficient while delivering superior performance.

## 4.5 ABLATION STUDY

**Results.** Table 5 details an ablation study on SelfMIS, quantifying the impact of removing the alignment mechanism and pre-trained weights. When the alignment mechanism is removed and the single-lead ECG model is directly trained in a supervised manner, the average macro AUC drops to 65.68, which is the lowest among all settings. Discarding pre-trained weights degraded performance: removal from the single-lead ECG model reduced macro AUC to 79.54 and reduced R@1 to 34.10. Conversely, removal from the multiple-lead ECG encoder decreased macro AUC to 76.71 and reduced R@1 to 11.94. Removing all pre-trained weights further lowered macro AUC to 76.56 and 8.68.

**Discussions.** The ablation study reveals that the alignment mechanism is the most critical component of the SelfMIS framework, as its removal causes a catastrophic decline for the single-lead ECG model. In comparison, the pre-trained weights of the multiple-lead ECG encoder are more critical than those of the single-lead ECG encoder; removing them leads to a significant drop in both retrieval and diagnostic performance. This is because, fundamentally, the single-lead diagnostic capability of SelfMIS is largely derived from the multiple-lead ECG encoder. The SelfMIS framework leverages the alignment mechanism to compel the single-lead ECG encoder to align its more limited input into this pre-defined, high-quality embedding space, effectively transferring diagnostic knowledge from a data-rich modality to a data-sparse one. Interestingly, even when trained entirely from scratch without pre-trained weights, SelfMIS still achieves respectable performance in myocardial infarction diagnosis. This suggests that enforcing representational consistency between single-lead and multi-lead views of the same cardiac event is sufficient to drive the single-lead ECG encoder toward learning an effective discriminative feature space for pathology. In summary, all components of SelfMIS contribute to the performance of model.

## 5 CONCLUSION

In this paper, we propose SelfMIS, a simple yet effective alignment learning framework to improve myocardial infarction detection from single-lead ECG. SelfMIS discards conventional manual data augmentation and instead directly aligns single-lead with multiple-lead ECG in the latent space by the self-cutting strategy, driving the single-lead ECG encoder to learn representations capable of inferring global cardiac context from local signals. This design provides richer and more informative representations for the single-lead ECG encoder, thereby enhancing the detection performance of myocardial infarction. Experimental results demonstrate that SelfMIS achieves superior performance while maintaining lower architectural complexity and computational cost compared to existing methods, thereby proving the effectiveness of direct alignment in the latent space. We hope that our approach will further inspire the community, particularly toward advancing ECG diagnostic performance on mobile devices, which are becoming increasingly prevalent.

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

# A    APPENDIX

## A.1    STATEMENT ON THE USE OF LLM

During the research and writing of this paper, we employed Large Language Models (LLMs) for auxiliary tasks such as generating short scripts during coding, as well as providing assistance in text translation and language polishing. Nevertheless, all core ideas, research methodologies, and academic perspectives were conceived and written independently by the authors, with the role of LLMs confined to enhancing the fluency and readability of the presentation.

## A.2    RELATED WORK

**Algorithms for Myocardial Infarction Detection.** The application of algorithmic techniques for automatic myocardial infarction detection constitutes a significant branch of AI for Healthcare (Han et al., 2024). Early research in this domain predominantly utilized traditional machine learning algorithms for diagnostic purposes. These approaches required the manual extraction of features—such as time-domain, frequency-domain, or various statistical features—from ECG signals, followed by classification using models like SVM (Dohare et al., 2018), KNN (Acharya et al., 2016) or DT (Fatimah et al., 2021). However, such a heavy reliance on feature engineering constrains their generalization to real-world clinical scenarios. Moreover, many of these methods perform downstream validation at a coarse granularity (e.g., simply differentiating between healthy and myocardial infarction patients) and seldom focus on localizing the infarction (e.g., detecting an anterior myocardial infarction). In recent years, deep learning algorithms have demonstrated notable advancements in myocardial infarction detection, often leveraging supervised learning with CNN (Pan et al., 2022), LSTM (Xiong et al., 2023), or Transformer (Yao et al., 2023) architectures. Nevertheless, the majority of existing methods are fundamentally based on multiple-lead ECG, with little exploration into the use of single-lead ECG. Moreover, these methods are generally not open-sourced, which limits their reproducibility and accessibility. Therefore, we select the recent ECGFounder as a representative supervised learning approach for comparison.

**Synthesis of Multiple-Lead ECG from Reduced Leads.** Synthesizing standard 12-lead ECG from limited leads has emerged as an important research direction in recent years. Early approaches typically relied on 2–3 leads (e.g., leads I, II, and V2) to infer the remaining leads, with representative implementations based on multilayer perceptrons (MLPs) Atoui et al. (2004) and hybrid convolutional neural network (CNN)–long short-term memory (LSTM) models (Gundlapalle & Acharyya, 2022). Subsequently, generative models were introduced, such as GAN-based architectures for synthesizing complete 12-lead ECG from a single lead (Seo et al., 2022), followed by EKGAN (Joo et al., 2023), which refined the GAN framework to better capture single-lead features. MCMA (Chen et al., 2024) proposed a multi-channel masked autoencoder capable of reconstructing 12-lead ECG from any single lead. Nevertheless, most of these methods remain optimized primarily at the signal level, leaving a substantial gap between the distributions of generated samples and real data in the latent space.

**Alignment Learning for ECG Signals.** In recent years, ECG alignment has been primarily achieved through contrastive learning, which has demonstrated a remarkable capability to learn generalized representations from unlabeled ECG signals and deliver substantial performance gains on downstream tasks (Lai et al., 2023). 3KG (Gopal et al., 2021) enhances the effectiveness of contrastive learning by employing physiologically inspired data augmentations, whereas CLOCS (Kiyasseh et al., 2021) leverages cross-spatial, temporal, and patient-level correlations. W2V-CMSC (Oh et al., 2022) further improves downstream adaptability by proposing a lead-agnostic self-supervised contrastive learning method. ISL (Lan et al., 2022) enhances cross-subject generalization through both inter-subject and intra-subject contrastive learning, while BTFS (Yang & Hong, 2022) improves ECG signal classification by integrating contrastive learning in both the time and frequency domains. However, existing methods still suffer from semantic corruption caused by data augmentation and a mismatch between invariance learning and the requirements of single-lead myocardial infarction detection, which ultimately leads to a performance drop.

## A.3 ABBREVIATIONS OF MYOCARDIAL INFARCTION TYPES

To ensure clarity throughout the paper, we summarize the abbreviations used for different types of myocardial infarction along with their corresponding full names in Table 6.

Table 6: Abbreviations of Myocardial Infarction Types.

| Abbreviation | Full Name |
|---|---|
| ALMI | Anterolateral Myocardial Infarction |
| AMI | Anterior Myocardial Infarction |
| ASMI | Anteroseptal Myocardial Infarction |
| ILMI | Inferolateral Myocardial Infarction |
| IMI | Inferior Myocardial Infarction |
| IPLMI | Inferoposterolateral Myocardial Infarction |
| IPMI | Inferoposterior Myocardial Infarction |
| LMI | Lateral Myocardial Infarction |
| PMI | Posterior Myocardial Infarction |

## B HYPERPARAMETER SETTINGS

To ensure reproducibility of our experiments, we summarize the hyperparameters used in both the pre-training stage and the downstream tasks. Table 7 lists the hyperparameter configuration for pre-training the model, while Table 8 presents the settings employed for fine-tuning on downstream tasks. These settings were determined based on empirical validation and follow common practices in related literature.

Table 7: Hyperparameters for Pre-training.

| Hyperparameter | Value |
|---|---|
| Batch size | 128 |
| Total epochs | 20 |
| Peak learning rate | $1 \times 10^{-4}$ |
| Weight decay | 0.1 |
| Optimizer | AdamW |
| Adam $\beta$s | (0.9, 0.999) |
| Adam $\epsilon$ | $1 \times 10^{-8}$ |
| Learning rate scheduler | Cosine |
| Gradient clip norm | 1.0 |
| Precision | AMP |

Table 8: Hyperparameters for Downstream Tasks.

| Hyperparameter | Value |
|---|---|
| Batch size | 64 |
| Total epochs | 20 |
| Peak learning rate | $5 \times 10^{-4}$ |
| Optimizer | Adam |
| Adam $\beta$s | (0.9, 0.999) |
| Adam $\epsilon$ | $1 \times 10^{-8}$ |
| Learning rate scheduler | Cosine |
| Patience | 5 |

