# OpenReview forum: "Self-Alignment Learning to Improve Myocardial Infarction Detection from Single-Lead ECG"
_ICLR.cc/2026/Conference — ICLR 2026 Conference Withdrawn Submission_

### Official Review · Reviewer_Pj64 · 2025-10-31

**Soundness:** 2
**Presentation:** 3
**Contribution:** 2
**Rating:** 2
**Confidence:** 4

**Summary:**

The paper introduces SelfMIS, a self-alignment pretraining framework for single-lead ECG myocardial infarction detection. The method forms “self-cut” positive pairs by pairing each multi-lead recording with its corresponding single-lead segment and trains a single-lead encoder to match the latent space of a frozen multi-lead encoder via a stop-gradient design, thereby removing manual augmentations. The motivation is that single-lead signals are information-limited and lack cross-lead context, while augmentation-driven invariance learning can distort ECG semantics and fails to cultivate the required cross-lead extrapolation. The key contribution is to shift the objective from transformation invariance to representation enrichment, producing information-enriched single-lead embeddings that improve downstream MI detection performance

**Strengths:**

- Clear problem framing: Precisely identifies the mismatch between transformation invariance and the single-lead requirement for cross-lead contextual extrapolation.

- Distinct conceptual contribution: Shifts the objective from invariance learning to representation enrichment via self-alignment.

- Simple and general method: Architecture-agnostic and plug-and-play as a pretraining head for existing encoders with minimal changes.

**Weaknesses:**

- Fundamental limitation unaddressed. Single-lead ECG provides only one projection of cardiac electrical activity. The paper does not explain how the method overcomes this intrinsic information deficit, and the Introduction largely sidesteps it.

- Strong data dependency and narrow experimental scope. The approach requires paired multi-to-single-lead pretraining, which limits data availability. Experiments do not demonstrate that the proposed training closes the purported “latent-space gap”: while the latent space gap in Fig 1 for existing methods provide a strong motivation, can the authors add the latent space results for the proposed methods, to show that such gap is closed with the proposed method?

- Teacher quality bottleneck. A frozen multi-lead encoder can propagate its biases and errors to the single-lead student. No mitigation (e.g., teacher ensembling, confidence weighting, or teacher refinement) is provided.

- While a good number of baselines is included, they are not well introduced making it difficult to assess the relevance of these baselines.
.

- Limited novelty. The clinical task is conventional and narrowly scoped, and the alignment design closely follows established teacher–student or contrastive consistency schemes with stop-gradient. The work appears application-driven and incremental rather than introducing a new learning principle, objective, or architecture.

- Given the small margins of improvements in the mean, it is important to provide statistics from multiple runs in the results (e.g., Table 1 & 2)

**Questions:**

Provide some additional evidence of “information completion” from a single lead beyond task AUC. Examples could include:
- Predict held-out lead embeddings (in comparison to embedding from multi-lead data)
- Predict selected clinical features (e.g., ST-elevation vectors) from the single-lead embedding.
- Given a single-lead embedding, retrieve its paired multi-lead counterpart among distractors; report top-k accuracy.

Clarify the basic concepts underlying the current baselines used, and when appropriate, add comparison with  strong knowledge-distillation methods, masked-modeling/reconstruction approaches (e.g., MAE/MIM), and single-to-multi-lead reconstruction-then-classification pipelines, as well as large supervised baselines.

---

### Official Review · Reviewer_Kcsi · 2025-10-31

**Soundness:** 3
**Presentation:** 3
**Contribution:** 3
**Rating:** 6
**Confidence:** 4

**Summary:**

This paper introduces a novel self-supervised learning approach aimed at aligning the single lead (Lead I) ECG encoding with the 12-lead ECG encoding. This alignment strategy is then assessed for the important task of Myocardial Infarction detection using a single ECG lead. Such an approach would allow for an earlier detection of MI, thanks to the widespread usage of wearable devices (Apple Watch, …).
The proposed method is compared with other generative models or self-supervised techniques and the results seem to indicate nice levels of performance.

**Strengths:**

The paper tackles an important problem, which is an readier detection of MI.
The authors performed a wide range of experiments and comparisons for assessing the proposed technique.

**Weaknesses:**

The structure of the paper is bit strange with several successive results and discussions subsections in the main result section. Some of these subsections present results from experiments that were not even introduced beforehand.
The ablation study is to very convincing as it does not allow for the evaluation of each different methodological decision (loss, stop grad, and so on..)
The proposed technique does not compare with most recent ECG generative models (Diffusion based, S4 models (Alcaraz et al) or Denoising Diffusion (Bedin et al).
The choice os a single database for either pertaining and downstream task does not allow for the assessment of the generalizability of the proposed technique.

**Questions:**

1 Did the authors use other preprocessing apart from the replacement of missing values ?
2 Given the use of the MIMIC IV database (single population, small number of different ECG devices), does it not restrict the generalizability of the trained representations ?
3 Was the PTBXL database not use for the training of the ECGFounder?
4 Why did the authors not assess their linear probe on another external database (other than PTBXL)? The 2023 CinC/PhysioNet challenge included other database with MI annotations (maybe not with the same fine -grain like localisation of the MI)
5 Would it be really useful to be able to localise the origin of MI with such a screening tool (one-lead ECG) ro would it not be enough to detect MI (without localizing its origin)? If so did the authors assess and compare their approach as a « simple » MI detector (I would imagine that other self-supervised technique could have higher levels of performance)
6 Did the authors envision to add a generative loss (or path) in order to learn an aligned encoding capable, still able to reconstruct « artificial » leads? If not, why do the authors think could help learning a better representation?

---

### Official Review · Reviewer_Fd5D · 2025-10-31

**Soundness:** 3
**Presentation:** 3
**Contribution:** 2
**Rating:** 2
**Confidence:** 3

**Summary:**

This paper addresses the limited spatial information problem in using single-lead ECG to detect myocardial infarction. An alignment learning framework is proposed that 1) a self-cutting strategy is used to construct positive pairs of single-lead and multi-lead ECGs and 2) the alignment is implemented by freezing encoder parameters on multi-lead ECGs and updating the single-lead ECG encoder. Experiments were conducted on nine MI types compared to different baseline methods.

**Strengths:**

- The limited spatial information of single-lead ECG is critical, and the idea of aligning it with multi-lead ECG is important.
- Experiments considered various MI types and were well-discussed.

**Weaknesses:**

- The latent space and how the spatial information is processed need to be better explained.
- More details of the methodology need to be provided.
- The experiments need better justification.

**Questions:**

- The proposed method aims to solve the limited spatial information in single-lead ECG from the perspective of latent space alignment. However, it is not clear whether the spatial information or other pathological information is kept in the latent space. Also, the spatial information varies across different patients. How does the learned alignment generalize to different patients?
- Section 2.3 needs to be elaborated. Is the ECG encoder the same for single-lead and multi-lead ECGs? If so, how does the ECG encoder learn effective information on the single-lead ECG, given the limited information? These are not detailed in the paper and the referenced ECGFounder.
- Section 2.4: The number of positive and negative pairs is imbalanced in Eq 3, with negative samples dominating the loss term. How does the loss function emphasize the positive pairs? Also, how are $t$ and $b$ chosen in the loss function?
- Section 3: The motivation for choosing selected baseline models should be explained. By replacing the missing values with 0, the inaccuracy is introduced in the multi-lead ECG. Would it be better to interpolate the missing recordings?
- The improvement of the proposed method in Table 1 is marginal compared to existing baselines, which is different from the statement of “clear advantage”. Could the authors provide visual examples to explain the change in the metric numbers?
- The authors should consider adding experiments to present the alignment of the latent space of the proposed method and comparison baselines.

---

### Official Review · Reviewer_ZmRx · 2025-11-01

**Soundness:** 3
**Presentation:** 1
**Contribution:** 1
**Rating:** 2
**Confidence:** 5

**Summary:**

This paper presents an intriguing alignment learning framework for single-lead ECG MI detection, demonstrating promising performance and efficiency, achieving an AUC of 0.83 on *the PTB-XL* dataset and outperforming baseline SSL methods (Table 2).

**Strengths:**

1. **Addressing a Critical Problem with a New Approach**: The paper clearly identifies limitations in existing single-lead ECG-based myocardial infarction detection (lack of spatial information, latent space discrepancy in generative models, inefficiency of prior alignment methods). By proposing SelfMIS, a novel latent space alignment framework to tackle these challenges, the authors highlight the value of their research. The shift in learning objective to "infer global cardiac context from local signals" is particularly innovative.

2. **Simple and Efficient Design**: A significant advantage of SelfMIS is its ability to achieve superior performance with a minimalist design, avoiding complex architectures or manual data augmentations. Its reliance on a "self-cutting" strategy and direct latent space alignment makes it practical and computationally efficient (refer to Table 4).

**Weaknesses:**

1. **Lack of Clarity and Independence in Figures and Tables:**

* Figure 1: The accompanying description is incomplete, and the left and right panels lack independent explanations, making it difficult for a reader to understand the intended message solely from the figure.
* Figure 2: Symbols such as $F_e$, $F_s$, and $F_m$ are not clearly defined in the caption, forcing the reader to search the main text. It is also not immediately clear from the figure alone that the red arrow in panel (c) signifies a stop gradient operation.
* Tables 1, 2, 3: The abbreviations for myocardial infarction types (e.g., ALMI, AMI) are not sufficiently explained within the table captions or the main body, requiring readers to consult the appendix. This diminishes the independence of the tables.
* In its current form, the paper’s readability and figure self-containment fall well below conference standards—captions are incomplete, key symbols undefined, and several figures cannot be interpreted without the main text, substantially limiting accessibility.

2. **Inadequate Consideration of the Target Audience in the Introduction:** The Introduction section dives into clinical details too quickly, potentially making it challenging for readers from machine learning/AI backgrounds to follow from the outset. Tailoring the introduction to the conference's audience is crucial.

3. **Questionable Motivation:**

* I'm not sure that generating other leads for training is the most intuitive idea. While converting a single-lead ECG into a multiple-lead ECG might seem intuitive to address the limited spatial information of a single-lead ECG, presenting it as "the most intuitive idea" might not resonate with all readers. Perhaps, it could be framed by contrasting it with other intuitive approaches like multi-task learning or different self-supervised representation learning methods.
* The paper's proposed method is distinctly different from generative models, yet the rationale for extensive comparison with generative methods in experiments is not clearly articulated. The paper mentions generative models' latent space gap and SelfMIS's latent space alignment to bridge this, but a more explicit connection between the motivation and the experimental design is needed to be truly convincing.

4. **Narrow Disease Scope and Justification:** The paper specifically targets Myocardial Infarction. While critical, the inherent challenge of single-lead ECG lies in its limited spatial information, which applies to various cardiac pathologies that might manifest differently across leads. If the goal was to highlight diseases specifically problematic for single-lead detection due to their localization, other conditions not well-captured by Lead I could have been considered, or the rationale for exclusively focusing on MI should be better justified.

5. **Potential Lack of Novelty Perception:** The approach might be perceived as an application of existing contrastive learning-based alignment methods like CLOCS. The authors claim SelfMIS focuses on "representation enrichment" rather than "transformation invariance." However, a more explicit explanation of how this differs fundamentally from CLOCS's cross-spatial alignment (learning similarities across different leads) is required to establish stronger novelty.

6. **Empirical rigor**
- Patient-level splits & leakage control:
  Given that the PTB-XL dataset already provides predefined folds split at the patient level, using these official folds is the appropriate choice to ensure patient-level independence and prevent data leakage.

- Multi-downstream evaluation:
  The study evaluates the proposed framework on a single downstream dataset. Demonstrating performance consistency across multiple downstream ECG tasks or datasets would strengthen generalizability and the claim of representation transferability.

- Statistical significance and reproducibility:
  The reported results appear to be based on single-run experiments without confidence intervals or measures of variability. Including confidence intervals (e.g., via bootstrapping or repeated runs with different seeds) would help quantify statistical reliability.

- Reproducibility:
  Clear disclosure of code, data usage, random seeds, hyperparameters, and compute budget would further improve empirical transparency.

**Questions:**

1. **Inclusion of Lead I in Multiple-lead ECG $\tilde{M}$:**

* In Methodology Section 2.2, $\tilde{M}$ is defined as having c leads, and  $\tilde{S}$ (Lead I) is extracted from  $\tilde{M}$ using self-cutting. However, Figure 2 (c) visually suggests that M input to f_m might be the remaining 11 leads, excluding Lead I.

* **Question:** Does the Multiple-lead ECG $\tilde{M}$  include Lead I, or is it composed of the remaining 11 leads? If it includes Lead I (implying $\tilde{M}$  is a 12-lead ECG and f_m processes the entire 12-lead ECG), then the representation of f_m's input in Figure 2 (c) without Lead I could be misleading. Clarification is needed. (Refer to Line 186-189 and Figure 2(c) in the paper).

2. **Negative Pair Construction in the Loss Function:**

* Equation (3) states that negative pairs are formed by combining single-lead ECGs with non-matching multiple-lead ECGs within the same batch. This aligns with instance-wise contrastive learning where $z_{ij}$ = -1 if $S_i$ and $M_j$ originate from different records.

* **Question:** In Figure 3, the Loss Function box's illustration of Negative Pairs might be misinterpreted as distances between representations of S and another S, or M and another M, rather than S and non-matching M. Could the authors clarify if the Figure 3 illustration of negative pairs is consistent with the definition in Equation (3)? Since SigLIP loss performs binary classification over all possible ($S_i$, $M_j$) combinations, intra-batch negative sampling is implied, and this should be explicitly clearer in the figure.

3. **Rationale for Performance Breakdown by MI Type:**
* **Question:** The paper presents performance results separately for each MI type (e.g., ALMI, AMI). What is the clinical or technical significance of this detailed breakdown? Beyond merely demonstrating performance across various types, a deeper analysis explaining why SelfMIS excels (or struggles) with specific types would significantly enhance understanding for readers. For example, what latent feature learning mechanism contributes to the remarkably high AUC for PMI (Tables 1, 2)?

4. **Improved Abbreviation Practice:**
* **Question:** For clarity, it would be beneficial to spell out abbreviations like AUC (e.g., Area Under the Receiver Operating Characteristic Curve (AUC)) or ALMI the first time they appear in the main text. While Appendix A.3 provides a table of MI type abbreviations, explicitly defining them upon first use in the main body would improve readability and prevent readers from having to constantly refer to the appendix.

---

### Note · Authors · 2025-11-14

I have read and agree with the venue's withdrawal policy on behalf of myself and my co-authors.